# The SWine IMputation (SWIM) haplotype reference panel enables nucleotide resolution genetic mapping in pigs

Rongrong Ding[1,2,3], Rodrigo Savegnago[2,10], Jinding Liu[2,4], Nanye Long[5], Cheng Tan[3,6], Gengyuan Cai[1,6], Zhanwei Zhuang[1], Jie Wu[1], Ming Yang[1], Yibin Qiu[1], Donglin Ruan[1], Jianping Quan[1,2], Enqin Zheng[1], Huaqiang Yang [1], Zicong Li[1,7], Suxu Tan[2,11], Mohammed Bedhane[2], Robert Schnabel [8], Juan Steibel[2,9], Cedric Gondro[2], Jie Yang [1,7✉], Wen Huang [2✉] & Zhenfang Wu [1,3✉]

Genetic mapping to identify genes and alleles associated with or causing economically important quantitative trait variation in livestock animals such as pigs is a major goal in animal genetic improvement. Despite recent advances in high-throughput genotyping technologies, the resolution of genetic mapping in pigs remains poor due in part to the low density of genotyped variant sites. In this study, we overcame this limitation by developing a reference haplotype panel for pigs based on 2259 whole genome-sequenced animals representing 44 pig breeds. We evaluated software combinations and breed composition to optimize the imputation procedure and achieved an average concordance rate in excess of 96%, a non-reference concordance rate of 88%, and an $r^2$ of 0.85. We demonstrated in two case studies that genotype imputation using this resource can dramatically improve the resolution of genetic mapping. A public web server has been developed to allow the pig genetics community to fully utilize this resource. We expect this resource to facilitate genetic mapping and accelerate genetic improvement in pigs.

[1] College of Animal Science and National Engineering Research Center for Breeding Swine Industry, South China Agricultural University, Guangzhou, Guangdong, China. [2] Department of Animal Science, Michigan State University, East Lansing, Michigan, USA. [3] Yunfu Subcenter of Guangdong Laboratory for Lingnan Modern Agriculture, Yufu, Guandong, China. [4] Academy for Advanced Interdisciplinary Studies, Nanjing Agricultural University, Nanjing, Jiangsu, China. [5] Institute for Cyber-Enabled Research, Michigan State University, East Lansing, Michigan, USA. [6] Guangdong Zhongxin Breeding Technology Co., Ltd, Guangzhou, Guangdong, China. [7] Guangdong Provincial Key Laboratory of Agro-animal Genomics and Molecular Breeding, South China Agricultural University, Guangzhou, Guangdong, China. [8] Division of Animal Sciences, University of Missouri, Columbia, Missouri, USA. [9] Department of Fisheries and Wildlife, Michigan State University, East Lansing, Michigan, USA. [10] Present address: Genus IntelliGen Technologies, De Forest, Wisconsin, USA. [11] Present address: College of Life Sciences, Qingdao University, Qingdao, Shandong, China. ✉email: jieyang2012@hotmail.com; huangw53@msu.edu; wzfemail@163.com

The domestic pig (*Sus scrofa*) is an important livestock species and a model organism for biomedical research[1]. Historically, domestication and intense artificial selection have created many pig breeds that are genetically and phenotypically distinct from each other and from their wild relatives[2–4]. More recently, high-throughput DNA sequencing and genotyping technologies[5] have facilitated the genetic improvement of pigs. For example, hundreds of genome-wide association and quantitative trait locus (QTL) mapping studies have identified numerous genomic regions associated with various production, physiological, and behavioral phenotypes[6]. These studies are important for understanding the genetic and biological basis of economically and biomedically important traits such as growth[7], fertility[8], and disease resistance[9].

The resolution of genetic mapping in pigs remains poor due in part to the low density of single nucleotide polymorphism (SNP) genotyping arrays. One proven, cost-effective approach to overcome the limitation in resolution is through genotype imputation, leveraging linkage disequilibrium to infer genotypes at unobserved polymorphic loci[10]. With large haplotype reference panels created by whole-genome sequencing, imputation has the potential to provide sequence-level genotypes[11]. In livestock animals, where QTL identification and genetic prediction are two major goals, and linkage disequilibrium is extensive, sequence-level genotype imputation has been successfully applied with a relatively small number of reference haplotypes but decent accuracy[12, 13]. In pigs, in particular, at least two public imputation servers are available[14, 15]. However, they either contained a very limited number of animals in the reference panel[14] or lacked good representation from major commercial breeds[15], limiting their applications. In addition, although many studies have demonstrated improvement in mapping resolution[16] and genomic prediction accuracy[17], none of these can be publicly accessed.

In this study, we produced whole-genome sequence data from 1530 newly sequenced pigs and combined them with 729 additional animals from public databases to call variants and develop by far the largest and most diverse reference panel of haplotypes in pigs to date. This substantial increase in the number of available genomes allowed us to impute SNP array genotypes to whole genome sequences rapidly and accurately. We evaluated the accuracy of imputation and demonstrated the utility of this haplotype reference panel in genome-wide association mapping. We introduce a new public web server (swimgeno.org) where users may submit array genotypes and retrieve imputed whole-genome sequence-level genotypes. This resource will greatly improve access to high-accuracy genotype imputation, facilitating potentially nucleotide resolution genetic mapping in pigs.

## Results

### Development of a haplotype reference panel consisting of >2000 pig genomes.
We consolidated whole-genome sequence data from newly sequenced animals ($n = 1530$) and publicly available data (Supplementary Data 1 and 2) for a total of 2259 pigs, representing 44 different breeds (Supplementary Data 1). The majority of animals were Landrace ($n = 651$), Yorkshire ($n = 543$), and Duroc ($n = 485$), three major commercial breeds. The uniquely aligned sequence depth was approximately 12.86 X averaged across all animals (Supplementary Data 1). We called variants using the GATK pipeline and calibrated variant quality scores with known variant sets compiled from commercial SNP arrays. After filtering out variants of low quality and excessive heterozygosity and missingness, 47.86 M autosomal variants remained. Sub-sampling of animals indicated that the increase in the number of discovered variants quickly diminished (Fig. 1a). More than 95% of all variants could be recovered using only 1000 randomly selected animals.

Linkage disequilibrium (LD) between variants in this population was extensive but differed by breed (Fig. 1b). LD in wild boars declined more rapidly as the distance between variants increased than in domestic breeds, consistent with the high level of inbreeding among intensively selected domestic breeds (Fig. 1b). Genetic variation present in the pig genome separated breeds into distinct clusters that represented geographic differentiation (Fig. 1c, d). The first principal component of the genotypes separated Asian breeds and wild boars from their European counterparts, while the second separated Durocs from other breeds (Fig. 1c). Estimated ancestries of the breeds also indicated clearly separated clusters according to their geographical locations (Fig. 1d). Taken together, the diverse and rich genetic variation in the 2259 pig genomes included in this study provides a strong foundation for whole-genome imputation.

**Accuracy of genotype imputation.** We focused on the ~34 M autosomal variants (30,489,782 SNPs and 4,125,579 indels) segregating at a minor allele frequency (MAF) > 0.005 to construct the haplotype reference panel. To investigate factors that influence imputation accuracy, we considered different combinations of commonly used phasing and imputation software, including SHAPEIT4/IMPUTE5, Beagle5.2/Beagle5.2, and Eagle2.4/Minimac4. We defined imputation accuracy using three metrics, the overall concordance rate between imputed and observed genotypes, non-reference concordance rate summarizing accuracy for non-reference genotypes only, and squared correlation ($r^2$) between imputed and observed genotypes. We focused on Landrace as the target set because it has the largest number of animals in the dataset. We held out 100 Landrace pigs sequenced at high coverage (>15X) and compared observed genotypes with imputed genotypes starting from sequencing-based genotypes at sites on a 50 K SNP array (GeneSeek GGP). Regardless of breed composition in the haplotype reference panel of fixed size, SHAPEIT4/IMPUTE5 outperformed Beagle5.2/Beagle5.2 and Eagle2.4/Minimac4 in all three metrics (Fig. 2a–c). SHAPEIT4/IMPUTE5 was therefore chosen for all subsequent analyses.

In cattle, imputation using multi-breed reference panels appeared to be more accurate than using a single-breed panel[12,18]. However, multi-breed panels are confounded by larger sample sizes. We asked whether imputation using reference panels of the same size from a single breed and from a mixture of multiple breeds made a difference (Fig. 3a, compare L, DLY, and LO). This question was important as it informs whether to use a multi-breed or breed-specific reference panel to achieve optimal accuracy. We again considered 100 Landrace animals as the target set because of its relatively larger sample size. We found imputation accuracy measured by all three metrics to be remarkably similar (Fig. 3b–d) when the reference panel size was equal. Reference panel derived from the same breed as the target set had a very slight advantage (Fig. 3b–d). However, multi-breed panels are useful because reference from the same breed alone (but smaller sample size) was not able to achieve the same accuracy (Fig. 3, compare L-250 with others). Because the vast majority of Landrace pigs were from a single population, the imputation accuracy may not reflect a realistic scenario when new target sets are derived from other populations. We evaluated imputation accuracy using 550 animals as the reference set but 41 Landrace pigs from the SRA as the target set, thus representing a situation where the target sets are distant from the reference. Imputation accuracies were lower, and the multi-breed panel appeared to hold a small advantage (Supplementary Fig. 1). Expanding the reference panel to 2218 animals increased the accuracy substantially (Supplementary Fig. 2). The lower accuracies may be due to a combination of the small number of

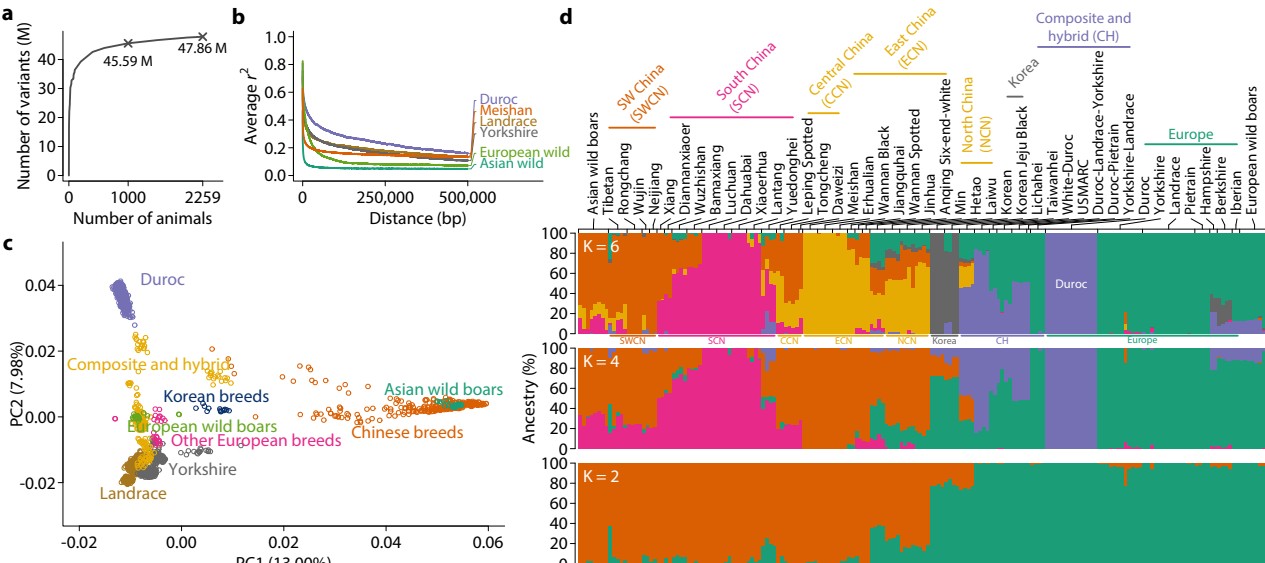

**Fig. 1 Genetic structure of the global pig population. a** Number of variants discovered as a function of the number of animals in the discovery cohort. The curve is generated by randomly subsetting the population and counting DNA variants that remain polymorphic. The number of variants discovered using 1000 and the whole set of 2259 animals are marked. **b** Pairwise linkage disequilibrium in four domestic breeds and wild boars from three regions. Average $r^2$ is plotted against the distance between variants. LD was calculated after removing low-frequency variants (MAF < 0.05) and close relatives (GRM > 0.5) in 435 Durocs, 522 Landraces, 493 Yorkshires, 36 Meishans, 24 European wild boars, and 27 Asian wild boars. **c** Scatter plot of first two principal components of genotype matrix for common (MAF > 0.05) and LD-pruned variants. Points are color-coded according to their reported breed information. A preliminary principal component analysis was performed to visually inspect and remove clear outliers from clusters, which indicated errors in breed information. **d** Ancestries of pigs were estimated with variable (K = 2, 4, 6) numbers of postulated ancestral populations using the ADMIXTURE software. Estimated ancestries were plotted as stacked bar charts with breeds annotated on the top. In addition to annotations above the bar chart, broad geographical locations are also annotated below the bar chart for K = 6.

target animals as well as further genetic distance from the reference panel. Taken together, although the comparison between multi-breed and breed-specific panels of the same size depends on specific situations, a multi-breed reference panel is desired as opposed to a breed-specific reference panel in most cases as it maximizes reference panel size.

We compared our SWine IMputation (SWIM) resource using the multi-breed reference panel with an imputation server for pigs (PHARP) that utilized 1006 animals publicly available in the SRA[15]. We evaluated imputation accuracy among variants that were present in both reference panels. PHARP contained relatively few major commercial breeds, including 115 Yorkshires, 85 Durocs, and 48 Landraces. We considered target sets from Landrace, Duroc, and Yorkshire, in which the vast majority of GWAS are conducted (Fig. 4a). When evaluating imputation accuracy, we held out 100 animals as the target set and used the remainder (n = 2159) as the haplotype reference panel. While the overall concordance rate was uniformly high (>94.24%), imputation using the SWIM panel developed in the present study was consistently higher than PHARP within each breed (Fig. 4b). The improvement was much more pronounced when considering the non-reference concordance rate and $r^2$, two metrics that more faithfully reflect the accuracy, especially at low frequency (Fig. 4c, d). The difference between SWIM and PHARP could simply be a sample size difference, especially for the breeds evaluated. The final reference haplotype panel consisting of all 2259 animals is expected to achieve a concordance rate in excess of 95.84%, a non-reference concordance rate of 88.26%, and an $r^2$ of 0.85.

We also assessed the performance of different starting SNP chips, including the GeneSeek GGP 50K, Affymetrix Wens 55K, and Affymetrix Axiom PigHD 660K. These chips were chosen because the Wens 55K and GGP 50K have a similar number of SNPs but share fewer SNPs, and the Axiom PigHD represents a

higher density. The imputation accuracies were evaluated in 100 Durocs and using 2159 animals as the reference (Supplementary Fig. 3a). After removal of SNPs whose probes did not map uniquely to the reference genome or were monomorphic, 39,491, 48,337, and 561,111 SNPs overlapped with the haplotype reference panel for the GeneSeek GGP, Wens, and Axiom PigHD, respectively (Supplementary Fig. 3b). As expected, higher density of SNPs led to higher imputation accuracy (Supplementary Fig. 3c–e) in all three metrics, with the Affymetrix PigHD 660K SNP chip achieving remarkably high accuracy at 99.50% overall concordance rate (Supplementary Fig. 3c), 98.63% non-reference concordance rate (Supplementary Fig. 3d), and 0.98 $r^2$ (Supplementary Fig. 3e).

**Genetic mapping using imputed sequence-level genotypes**. To demonstrate the usefulness of sequence-level genotype imputation in genetic mapping, we performed genome-wide association studies (GWAS) for two important growth traits in pigs, using both SNP arrays and imputed genotypes. The two traits, backfat thickness and body length, were chosen because putative causal genes and mutations have been previously well characterized. Our objective was to see if imputation-based GWAS was able to find previously validated functional genes and variants.

*Backfat thickness.* Backfat thickness (BF) is one of the most important economic traits in pigs and has been intensively interrogated for its genetic basis. Genomic heritabilities estimated using either array SNPs or imputed SNPs were similar and indicated a moderately heritable trait (Fig. 5a). Alleles in several genes, including *IGF2*[19,20], *MC4R*[21], and *LEPR*[22], have been consistently associated with BF variation in pigs. In particular, a missense mutation in the *MC4R* gene (chr1:160773437:G>A) has

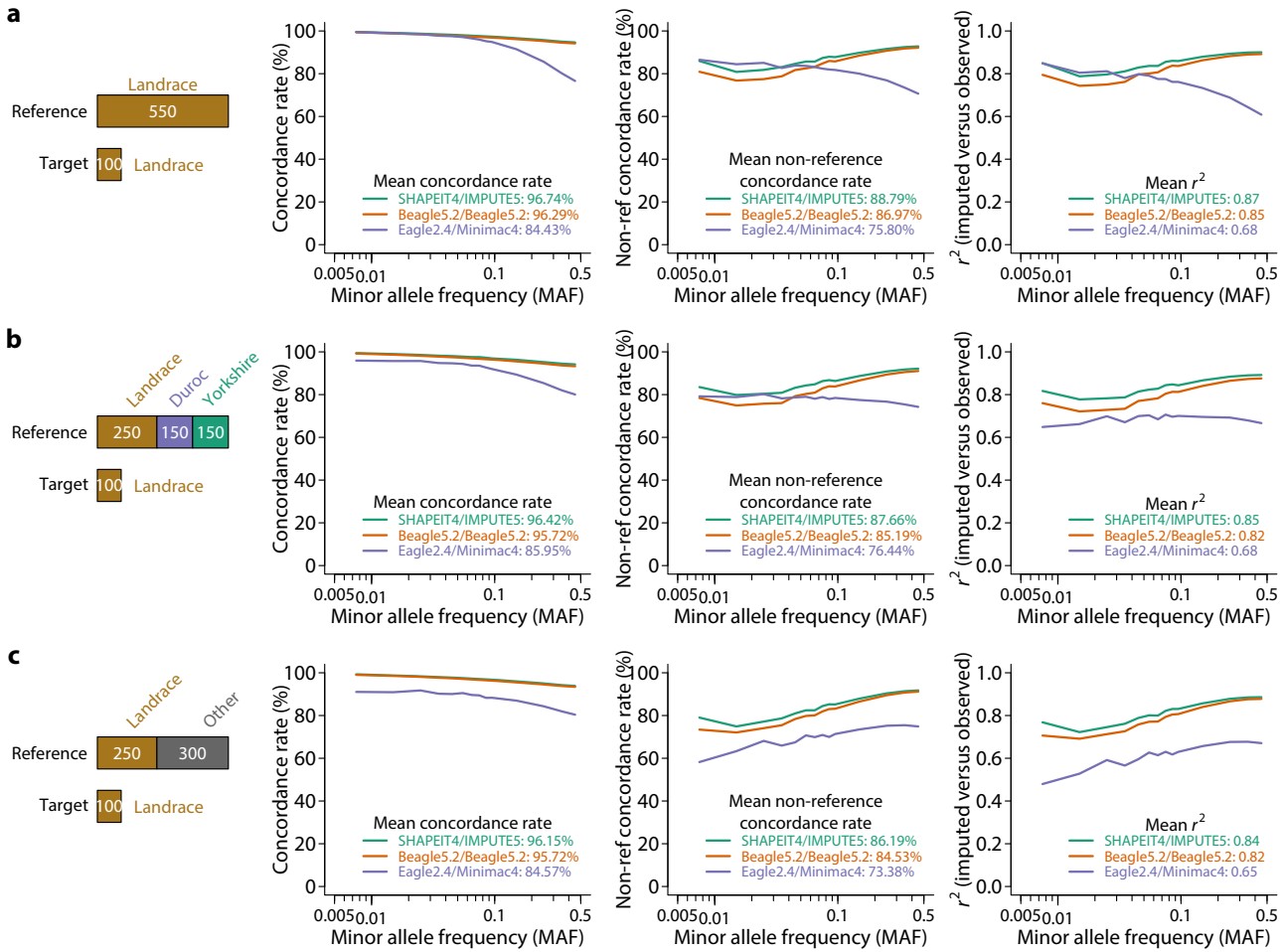

**Fig. 2 Comparison of software combinations for imputation. a** Concordance rate, non-reference concordance rate, and $r^2$ of imputed versus observed genotypes using different software combinations with 550 Landraces as the reference panel. **b** Same analysis but in a reference panel consisting of 250 Landraces, 150 Durocs, and 150 Yorkshires. **c** Same analysis but in a reference panel consisting of 250 Landraces and 300 other breeds (not Duroc or Yorkshire).

been suggested as the causative mutation[21] and extensively replicated in multiple genetic backgrounds[23]. Furthermore, mutations in *MC4R* are strongly associated with early onset obesity in humans[24], and its role in the regulation of energy homeostasis is well established[25]. Importantly, the putative causal mutation in *MC4R* has been included in one of the commercially available SNP genotyping arrays, the Geneseek GGP Porcine 50K SNP Chip (Neogen, Lincoln, NE). However, the same SNP is not present in the more widely used Illumina PorcineSNP60 chip. To see if genotype imputation was able to correctly impute the genotypes of this SNP, we excluded the *MC4R* SNP and imputed whole-genome genotypes from a population of 3769 Duroc pigs genotyped using the GGP Porcine 50K SNP arrays. Remarkably, the concordance rate and $r^2$ between the imputed and array *MC4R* SNP genotypes were 99.71% and 0.9916, respectively. We performed GWAS using array and imputed genotypes; both showed a major peak on chromosome 1 (Fig. 5a, Supplementary Data 3 and 4) and a clear deviation of *P*-value distribution from the null (Supplementary Fig. 4a). Using imputed genotypes, the highest hit from imputed SNPs (chr1:161511936:T > C, $P = 2.98 \times 10^{-13}$) explained 2.85% of the total phenotypic variance (Fig. 5a). Under this peak in a 4-Mb region (158.5–162.5 Mb), there were 7138 variants within 22 genes. Linkage disequilibrium in this region was extensive, with 1050 variants in strong LD ($r^2 > 0.8$) with the top hit, including the

*MC4R* SNP (Fig. 5b). The highest hit was an intronic SNP in the gene *CCBE1* (Fig. 5b). However, the extensive LD in this region makes it difficult to pinpoint a causative mutation by genetic data alone. Additional functional information and genetic data that break the LD are necessary to further fine-map causative genes and mutations. Nevertheless, the ability to identify the putative *MC4R* causative SNP as one of the top associated variants in a long stretch of high LD region clearly demonstrated the improvement of resolution using imputed genotypes. In our analysis, the *MC4R* SNP was initially removed and would otherwise be invisible without the imputation, as would be the case if the Illumina PorcineSNP60 chips were used.

*Body length*. We next considered body length. We imputed genotypes from an Affymetrix 55K SNP chip (Wens55K) to a whole genome sequence using our imputation platform and performed GWAS in a population of 1694 Yorkshire boars (Fig. 6a). The trait has a moderately high heritability, as estimated using both array ($h^2 \sim 0.32$) and imputed ($h^2 \sim 0.34$) genotypes (Fig. 6a). Using GWAS (Supplementary Fig. S4b), we found a highly significant peak on chromosome 17 (Fig. 6a, Supplementary Data 5 and 6) where the lead variant was an intergenic SNP upstream of the *BMP2* gene (chr17:15643342:C>T, $P = 3.45 \times 10^{-39}$). Remarkably, this variant explained 13.65% of the total phenotypic variance, and the homozygous C/C animals

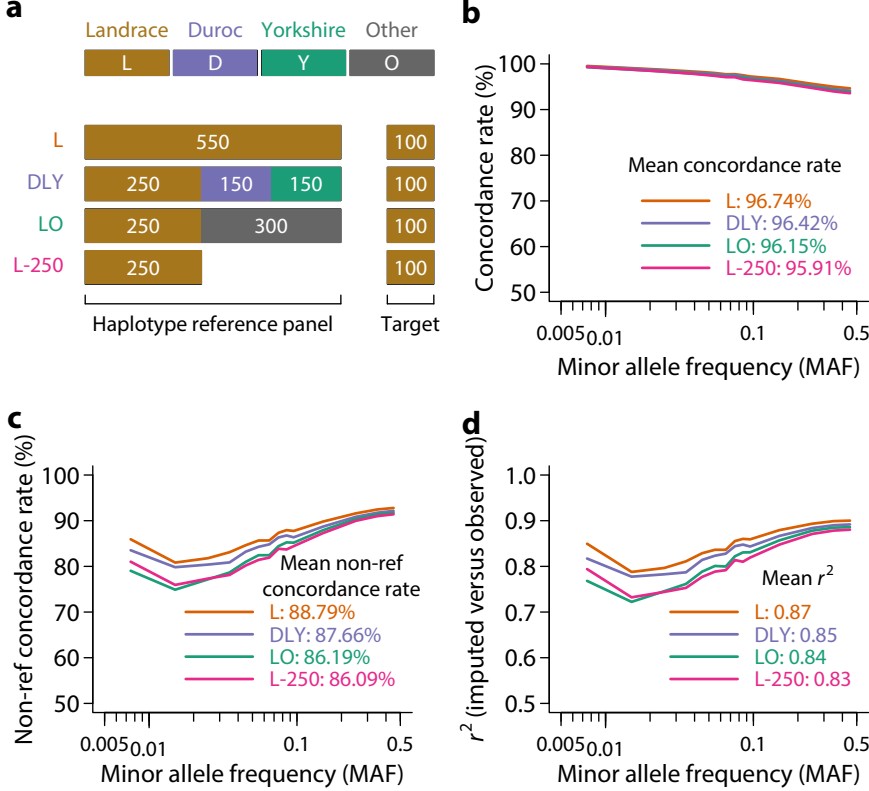

**Fig. 3 Effects of breed composition of haplotype reference panel on imputation accuracy. a** Experimental design to investigate the effect of breed composition of haplotype reference panel on imputation accuracy. Three reference panels were tested, including 'L': 550 Landrace animals; 'DLY': 550 pigs from the Duroc, Landrace, and Yorkshire breeds; 'LO': 550 pigs from Landraces and other non-Duroc or Yorkshire breeds; 'L-250': 250 Landrace animals only. One hundred Landraces were used as the target set. Concordance rate (**b**), non-reference concordance rate (**c**), and $r^2$ (**d**) of imputed versus observed genotypes using different breed compositions of the haplotype reference panel.

were, on average, 4.01 cm longer than the T/T homozygotes (Fig. 6b, c). *BMP2* has been repeatedly shown to be associated with growth traits in pigs. A recent study implicated a regulatory variant upstream of the *BMP2* gene and validated its functional impact using reporter genes[26]. This regulatory variant was the third most significant SNP under this peak in our analysis. Whether one or both of these potentially regulatory variants are the causative mutations remains to be determined. Given the strong association, high MAF of these SNPs, and less extensive LD in this region, it is unlikely that these regulatory variants were tagging protein-coding and less common variants in the *BMP2* gene. In addition to the genetic support from this Yorkshire population, the body length increasing C allele was much more prevalent in Landrace than in other breeds. A hallmark of the Landrace breed is its long body size; thus, regulatory variation of the *BMP2* gene may be a major contributor to the phenotypic differentiation between pig breeds. In contrast, although the SNP chip was able to broadly identify this region, the most significant SNP (chr17:15827832:T>G, $P = 1.58 \times 10^{-25}$) in an SNP chip-based GWAS was about 184 kb away from the lead SNP and explained a substantially smaller variance (8.22% versus 13.65%).

**SWine IMputation (SWIM) server**. To enable the broad research community to efficiently utilize the resource developed in this study, we developed a public SWine IMputation (SWIM) web server (https://www.swimgeno.org and https://swim.scau.pigselection.com/swim), on which users can upload SNP chip genotypes and retrieve imputed genotypes. The user interface is extremely simple, which only requires users to upload the genotypes in gzipped ped/map format and leave their email

addresses. Unlike other servers, such as PHARP, allele matching and flipping are performed on the server end, further simplifying the process on the user end. Imputation status can be monitored and results can be downloaded from a dynamic link without having to register an account. The server is set up to accommodate multiple users at the same time while limiting multiple jobs of the same user. Our tests indicated that a typical job with 2000 individuals and 50K SNP chip genotypes can be completed in approximately 12 h for all chromosomes.

**Discussion**

We present here the development of the largest reference haplotype panel in pigs and an accompanying web server for the public to utilize this resource for genotype imputation. The high level of diversity and the large number of animals in the panel enabled us to achieve very high imputation accuracy with concordance rate, non-reference concordance rate, and $r^2$ in excess of 95.84%, 88.26%, and 0.85, respectively, starting from 50K SNP arrays (Fig. 2). The accuracies were comparable to those obtained with medium density SNP arrays within pedigreed populations[27]. Given the high accuracy and easy access with no requirement for pedigree, we expect this public resource to vastly democratize sequence-level imputation in pigs and accelerate genetic discoveries. The SWIM server only supports SNP chip-based imputation at present. Low-coverage sequencing-based imputation is much more challenging to accommodate on a web server due to its requirement for massive computational resources. Nevertheless, users may implement their low-coverage sequencing-based imputation using the haplotype reference panel we share.

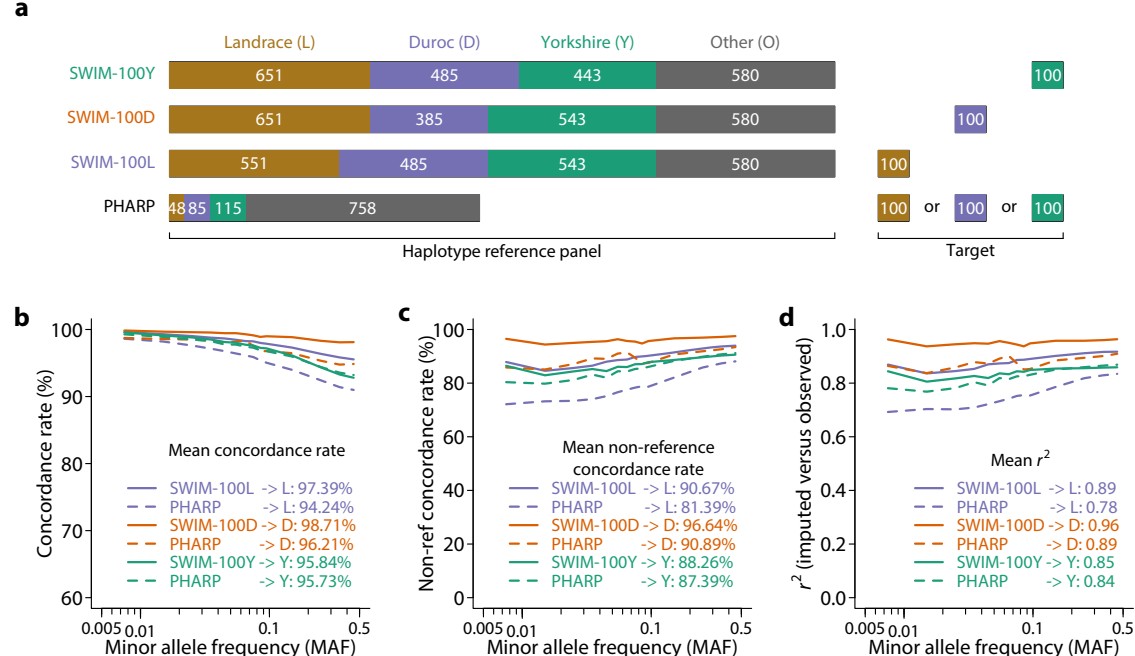

**Fig. 4 SWIM provides high imputation accuracy in pigs. a** Composition of the haplotype reference panels, including different scenarios of SWIM and PHARP, and target set. SWIM-100Y, SWIM-100D, and SWIM-100L are holding out 100 Yorkshires, 100 Durocs, and 100 Landraces as the target set, respectively. For the PHARP reference panel, the same 100 Yorkshires, 100 Durocs, and 100 Landraces are used to evaluate imputation accuracy. **b** Concordance rate of imputed versus observed genotypes using different haplotype reference panels. The mean concordance rate across all variants is also indicated on the plot for each reference panel. **c** Non-reference concordance rate of imputed versus observed genotypes using different haplotype reference panels. The mean non-reference concordance rate across all variants is also indicated on the plot for each reference panel. **d** $r^2$ of imputed versus observed genotypes using different haplotype reference panels. Mean $r^2$ across all variants is also indicated on the plot for each reference panel.

High-throughput genotyping arrays greatly simplified genotyping, and numerous new QTLs have been mapped by association mapping, typically within a breed and with hundreds to thousands of individuals[6]. However, while the resolution has improved with SNP arrays, causative genes and mutations remain extremely elusive, partly because SNP arrays prioritize assay feasibility, homogeneous spacing, and common SNPs[5].

Our evaluations indicated that Shapeit4/Impute5 outperformed other software combinations, higher density of SNP chips led to higher imputation accuracy, and multi-breed haplotype reference panels maximizing sample size were preferred. Importantly, animals that were genetically closer to the haplotype reference panel could be imputed with higher accuracy. This further reinforces the importance of data sharing to increase representation in the haplotype reference panel.

As we have shown with the examples above, imputation is expected to greatly improve the resolution of gene mapping. Given the large number of existing genome-wide association studies in pigs[6], we expect this resource to be highly utilized and impactful. Indeed, more than 130,000 genomes were imputed in the first year since the server became public, including a recent study that found SWIM imputed genomes to detect more significant SNPs compared to other platforms[28]. All existing studies using SNP arrays can be improved by a simple imputation followed by GWAS without additional data. Meta-analysis also becomes possible because a common SNP set can be obtained. Nonetheless, the resolution of genetic mapping depends not only on SNP density but also on experimental design and genetic structure in the mapping population. Sequence-level imputation does not necessarily identify causative mutations in one single step[16]. The availability of this resource will allow for suitable designs of mapping studies to achieve the highest possible resolution in specific circumstances and potentially nucleotide resolution.

## Methods

**WGS data collection**. We consolidated WGS data from multiple sources. A total of 1530 animals are first reported in this study using Illumina ($n = 863$) and BGI ($n = 667$) platforms with 150 bp paired-end reads. Among them, 610 Landrace, 413 Duroc, 391 Yorkshire, 18 Taiwanhei, and 17 Lichahei were from Wen's Food Group Co., Ltd. (Yunfu, Guangdong, China), 21 Dahuabai, 21 Lantanghei, 20 Guangdong Xiaoerhua, and 19 Yuedonghei from Guangdong Gene Bank of Livestock and Poultry (Guangzhou, Guangdong, China). Additionally, sequences for 729 animals were downloaded from the sequence read archive (SRA). A complete breakdown, including accession numbers, sample sizes, and average sequencing coverage, can be found in Supplementary Data 1 and 2.

**Variant calling, recalibration, and filtering**. We aligned sequence reads to the pig reference genome (Sscrofa11.1, a Duroc pig)[29] using BWA-MEM-0.7.17[30] and called variants (in GVCF format) using GATK-4.1.8.1 HaplotypeCaller[31] after several post-alignment processing steps including duplicate removal using PicardTools-2.23.3[31], and base quality recalibration using GATK. A population VCF was generated by combining GVCFs across all samples. Variants with excessive heterozygosity ("ExcessHet > 54.69") were removed. Variant quality score recalibration (VQSR) on SNPs was performed with truth SNP sets compiled from commercial SNP arrays, including 50K, 60K, and 80K SNP chips (prior = 15.0) on the Illumina platform and the 660K (prior = 12.0), SowPro90 (prior = 15.0) SNP chips from the Affymetrix platform. SNPs were filtered with a truth sensitivity filter level at 99.0. Without a truth set of indels, we applied hard filtering on them by excluding indels with QD < 2.0, QUAL < 50.0, FS > 100.0, ReadPosRankSum < −20.0, as recommended by GATK's best practices. Additionally, we filtered out animals with a missing rate >0.20, heterozygosity >0.20, and retained bi-allelic sites with a missing rate <0.2 and mean sequencing depth between 5 and 500. Filtering was performed using a combination of VCFtools 0.1.13[32] and BCFtools 1.13[33] commands.

**Population genetics analysis**. Linkage disequilibrium was computed using PopLDdecay[34] on individuals within the same breed after removing close relatives (GRM > 0.5) and low-frequency variants (MAF < 0.05). To understand the genetic structure in the population, we retained variants with MAF > 0.05 and missing rate <0.1 and pruned SNPs with LD ($r^2 < 0.3$, -indep-pairwise 50 10 0.3) using PLINK 1.9[35]. Principal component analysis (PCA) was performed on the filtered list of 1,223,882 variants using GCTA 1.93.2[36] for all individuals. Ancestries were estimated using ADMIXTURE 1.3[37] on 185 individuals randomly selected according

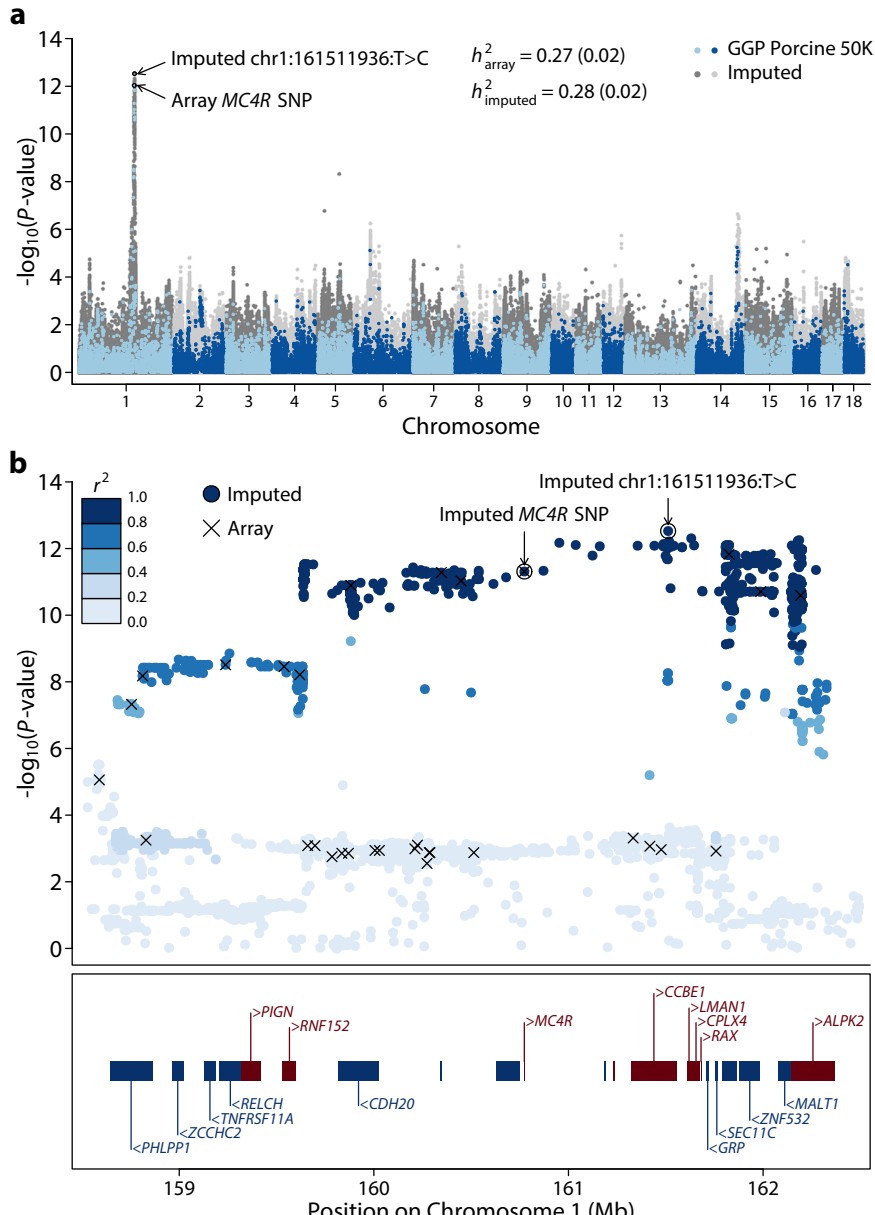

**Fig. 5 Genetic mapping of backfat thickness in pigs. a** Manhattan plot of genome-wide association studies (GWAS) for backfat thickness. The gray (dark and light) points on the background are from GWAS using imputed genotypes, while the blue (light and dark) points are from GWAS using SNP chips. Genomic heritabilities calculated using array and imputed genotypes are indicated. The most significant SNPs from GWAS using imputed and array genotypes are indicated by circles and arrows. **b** Association within the 158.5–162.5 Mb region of chromosome 1, where the top hits in GWAS are located. Points indicate $-\log_{10}(P\text{-value})$ along the chromosome using imputed genotypes and SNPs where arrays also have genotypes are marked by crosses. The top SNPs from GWAS using imputed and array genotypes are marked by circles and arrows. $r^2$ between the SNPs and the top SNP (chr1:161511936:T > C) is indicated by a gradient of blue color. Locations of genes are indicated in the box below the plot, where blue boxes and gene names with a left arrowhead (<) indicate genes transcribed on the reverse strand, and red boxes and gene names with a right arrowhead (>) indicate genes transcribed from the forward strand. Genes that are not marked do not have gene symbols. Gene locations are based on the Ensembl Release 98 annotation.

to breed representation in the dataset or at least four individuals per breed. The downsampling was necessary to properly visualize population structure.

**Genotype imputation**. We further filtered variants prior to phasing haplotypes in the reference population. Variants with missing rate >0.1 and MAF < 0.005 were removed. Additionally, variants with a Hardy–Weinberg equilibrium test $P$-value < $10^{-10}$ implemented separately in PLINK in all three of the Duroc, Landrace, and Yorkshire pigs were removed. Only autosomal variants were retained for imputation.

We extracted 100 Landrace pigs with the highest sequencing depth (17.42 X average sequencing depth, ranging from 14.98 to 63.11 X) and designated these individuals as the target population to evaluate imputation accuracy. To test the effect of breed composition of the reference population, we constructed four

reference haplotype panels using different sets of individuals, including All ($n = 2159$): all individuals except the 100 Landraces; L ($n = 550$): Landrace pigs only; DLY ($n = 550$): 250 Landraces + 150 Durocs + 150 Yorkshires; and LO ($n = 550$): 250 Landraces + 300 randomly selected pigs other than Durocs and Yorkshires. Phasing was independently performed in these reference sets. In addition, we also tested imputation using the PHARP web server (http://alphaindex.zju.edu.cn/PHARP/index.php), which contains reference haplotypes constructed from 1006 individuals in the SRA.

We tested three combinations of software for phasing and imputation, including SHAPEIT 4.2[38] + IMPUTE5 1.1.5[39], Beagle 5.2[40] + Beagle 5.2, and Eagle 2.4[41] + Minimac 4[42]. All software tools were run with default options and an uninformative linkage map (1 cM per 1 Mb), but the effective population size was set to 100. Imputed genotypes were called by the ones with the highest posterior

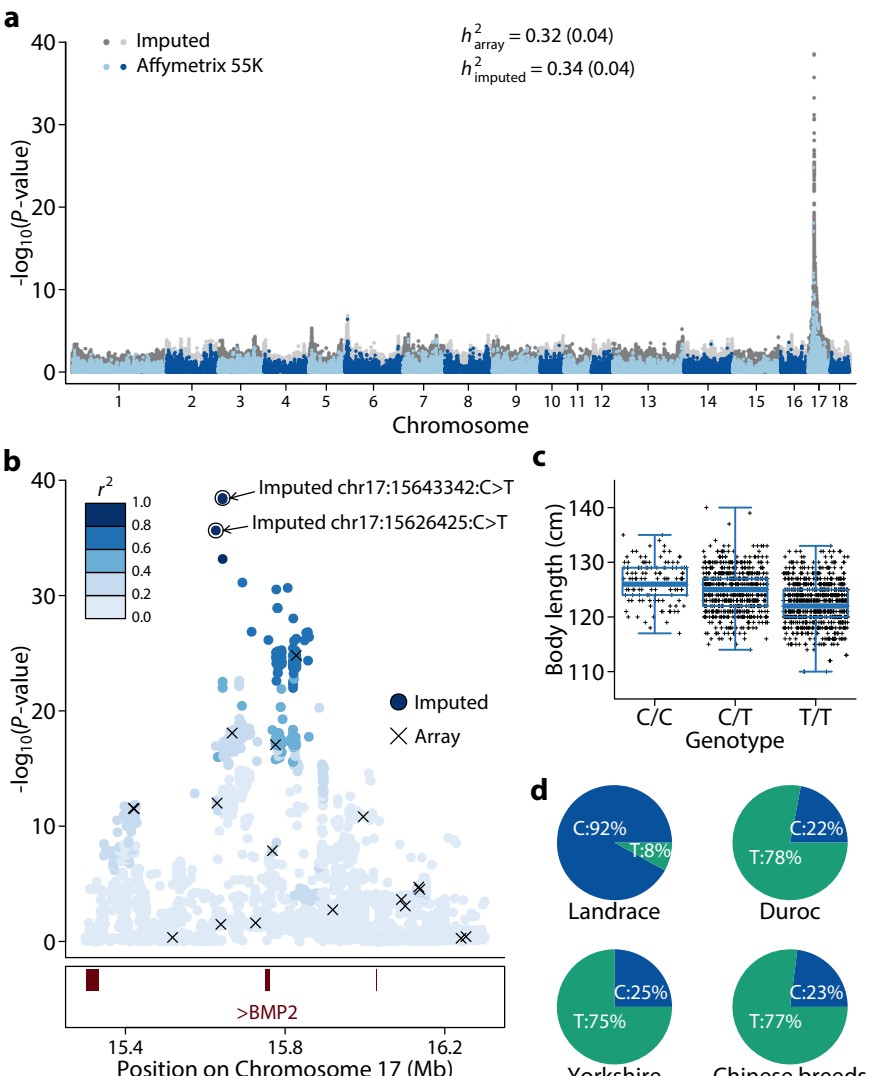

**Fig. 6 Genetic mapping of body length in pigs. a** Manhattan plot of genome-wide association studies (GWAS) for body length. The gray (dark and light) points on the background are from GWAS using imputed genotypes, while the blue (light and dark) points are from GWAS using SNP chips. Genomic heritabilities calculated using array and imputed genotypes are indicated. **b** Association within the 15.3–16.3 Mb region of chromosome 17, where the top hits in GWAS are located. Points indicate $-\log_{10}(P\text{-value})$ along the chromosome using imputed genotypes and SNPs where arrays also have genotypes are marked by crosses. The top SNPs from GWAS using imputed and array genotypes are marked by circles and arrows. $r^2$ between the SNPs and the top SNP (chr17:15643342:C>T) are indicated by a gradient of blue color. Locations of genes are indicated in the box below the plot and according to the Ensembl Release 98 annotation. All three genes are colored in red and transcribed from the forward strand. The only gene with a symbol in this region is BMP2. **c** Scatter and box plots of body length (in cm) for the three genotypes of the chr17:15643342:C>T SNP. The lower and upper boundaries of the box are, respectively, 25% and 75% quantiles of the data, the midline median, and the whiskers minimum and maximum. **d** Allele frequencies of the chr17:15643342:C>T SNP in different breeds.

genotype probability. However, users of the imputation web server also receive genotype probabilities.

We considered three commonly used metrics of imputation accuracy, concordance rate, non-reference concordance rate[43], and $r^2$. Concordance rate is defined as the proportion of individuals with imputed genotypes in concordance with observed genotypes. Non-reference concordance rate is similar to the concordance rate but is restricted to only individuals that are not homozygous for the reference allele. $r^2$ is the squared Pearson correlation coefficient between observed and imputed genotypes. We measured concordance rates and $r^2$ on a per SNP basis and averaged them over SNPs in MAF bins or across the whole genome.

**Genotypic and phenotypic data collection.** To demonstrate the utility of imputation in genetic mapping, we collected phenotypes and genotypes for three populations of pigs, which were managed by three core breeding farms of Wen's Food Group Co., Ltd. (Yunfu, Guangdong, China), all under standard management practices. For backfat thickness, the phenotypes were collected on 3769 Duroc pigs from 2013 to 2018, and SNP genotyping was performed using the Geneseek GGP Porcine 50K SNP chip (Neogen, Lincoln, NE, USA). Backfat thickness was

measured between the 10th and 11th ribs using an Aloka 500 V SSD B ultrasound (Corometrics Medical Systems, USA) when live weights of pigs reached about 100 kg (100 ± 5 kg). For body length, phenotypes from a total of 1694 Yorkshire boars were collected from 2012 to 2018, and SNP genotyping was performed using the Affymetrix PorcineWens55K SNP chip (Affymetrix, Santa Clara, CA, United States). Body length was measured from the base of the ear to the base of the tail in pigs at approximately 100 kg (100 ± 5 kg) body weight. All samples were collected according to the guidelines for the care and use of experimental animals approved by the Ministry of Agriculture and Rural Affairs of the People's Republic of China. The ethics committee of South China Agricultural University specifically approved the animal use in this study.

**Genome-wide association studies.** We used GCTA 1.92.1 to perform a mixed linear model (MLM) based association analysis. The following statistical model was used: $y = \mu + xb + g + e$ (Equation 1), where y is the vector of the phenotypic values for all animals, $\mu$ is the intercept, $x$ is the design matrix coding genotypes and other incidences of fixed effects, $b$ is the vector of fixed effects including SNP effect and additional covariates such as sex, pen, year-season effects depending on

the traits, and $g$ is the vector of polygenic random effects with covariance dictated by the genomic relationship matrix, and $e$ is the vector of random residuals. We used SNPs on the GeneSeek GGP 50 K SNP chip (for backfat thickness) and Affymetrix Wens 55K SNP chip (for body length) to compute the genomic relationship matrix. We used a genome-wide significance threshold of $P = 5 \times 10^{-8}$ to declare significance. Variance explained by a single significant SNP was estimated by fitting a mixed linear model with the genomic relationship matrix determined by a single SNP.

**Statistics and reproducibility**. All statistical analyses are performed using either software packages as described or in R 4.2.2. We supply all scripts, including those to generate figures in a GitHub (https://github.com/qgg-lab/swim-public) as well as a Zenodo repository[44] (https://doi.org/10.5281/zenodo.7900470). The sample size for the entire SWIM haplotype reference panel is 2259, with subsets selected for the different designs to answer specific questions. Sample sizes for the backfat thickness and body length GWAS were 3769 and 1694, respectively.

**Reporting summary**. Further information on research design is available in the Nature Portfolio Reporting Summary linked to this article.

## Data availability

Raw sequence data for 512 animals have been deposited to SRA (PRJNA842867). Additional sequenced animals were proprietary properties of Wen's Food Group Co., Ltd. and Guangdong Gene Bank of Livestock and Poultry. They may be requested by contacting research-pig@wens.com.cn and yangh@scau.edu.cn, respectively. Raw sequence data for a subset of the animals ($n = 729$) utilized in this study were downloaded from SRA (Supplementary Data 1 and 2). Imputation utilizing the full dataset is delivered as a web service (https://www.swimgeno.org and https://swim.scau.pigselection.com/swim) and is publicly available. Phased haplotypes from all publicly available individuals, including this study ($n = 1241$), are available as VCF files at https://quantgenet.msu.edu/swim/statistics.php. Source data underlying Figs. 1a, b, 2, 3, 4, and 6c are provided in Supplementary Data 7, 8, 9, 10, 11, and 12, respectively.

## Code availability

All computer codes, including all analyses performed in this study and codes for the SWIM web server, are available at https://github.com/qgg-lab/swim-public and at a Zenodo repository[44] (https://doi.org/10.5281/zenodo.7900470).

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

## Acknowledgements
This work is supported by a USDA-NIFA project (2021-67021-34149 to W.H., C.G., J.S., and R.Sc.), a USDA-NIFA Hatch project (MICL 02560 to W.H.), a Natural Science Foundation of China project (31972540 to J.Y.), a Natural Science Foundation of Guangdong Province project (2018B030313011 to Z.W.), and a Key Technologies R&D Program of Guangdong Province project (2022B0202090002 to Z.W.). The web server (https://www.swimgeno.org) is supported by the USDA Swine Genome Coordinator Fund (NRSP8).

## Author contributions
W.H., Z.W., J.Y., and R.D.: conceptualization and design; R.D., R.Sa., N.L., and W.H.: developed and optimized pipeline; R.D., S.T., and M.B.: analyzed data; J.L. and W.H.: developed web server; R.Sc., C.T., G.C., Z.Z., J.W., M.Y., Y.Q., D.R., J.Q., E.Z., H.Y., Z.L., J.S., and C.G.: contributed tools and data; R.D. and W.H.: wrote the paper, with input from all authors.

## Competing interests
C.T. and G.C. are employees of the Guangdong Zhongxin Breeding Technology Co., Ltd. All other authors declare no competing interests.
