## [Peer Review File · Communications Biology]

Reviewers' comments:

Reviewer #1 (Remarks to the Author):

I thought the manuscript was generally well written with nice clear figures. My comments in no particular order are below.

R2 is correlated to allele frequency and Fig 1b seems to be assuming that the distribution of allele frequencies is the same for each breed. This may not be the case? For example there may be more rare variants in certain populations impacting these results which should be checked? Also is the number of animals in each group the same?

Also in the legend of Figure 1b the authors say "LD was calculated with common variants (MAF>0.05) and close relatives (GRM>0.5) removed" but in methods (line 261) say "individuals within the same breed with close relatives (GRM > 0.5) and low frequency variants (MAF < 0.05) removed". Which seem to contradict each other with respect to whether only rare variants were kept or excluded.

Figure 1D, should say "annotated above the barchart"

Nomally x axis of Figure 2/3 plots would be on a log scale, as differences are generally at the lower allele frequencies (which also generally contains most variants) e.g. see <https://www.nature.com/articles/s41588-020-00756-0/figures/2>

For analysis in Figure 2 were the authors sampling the 50K SNP genotypes from the WGS data or did they have matching array data for these samples? If the former I assume not every genotype was called in the WGS data following filtering? If so should state what proportion of array genotypes were not called in the WGS data and what number were left to do the imputation with.

Is it worth comparing imputation performance from variants on some of the other pig arrays? For example the 650k axiom array? At least for the four arrays the authors say are supported by their server here: <https://quantgenet.msu.edu/swim/statistics.php>
On line 214 they say Figure 2 was based on a 60K array but on line 109 they say was a 50K array.

The authors suggest that for a fixed size a reference panel derived from a matching breed performs better than one composed of multiple breeds. This is contrary to the hypothesis that multi-breed reference panels better represent haplotypes that may be rare in a particular breed, improving general imputation performance. How well though does this analysis reflect likely downstream usage as 610 of the 651 landrace genomes used in this study were obtained from the same company and are therefore likely unusually related to one another. So this represents a specific sub-population of landrace animals rather than landrace as a whole. Does the imputation performance drop in the landrace animals from other cohorts?

Why is there an uptick in performance accuracies at the lowest allele frequencies in Figures 2 and 3? Normally the rarer the variant the less well imputed they are.

Resource spelt wrong on line 126

Can the authors make the genotypes callset (VCF) for at least the 729 previously publically available and 512 new genomes available? This will likely be of substantial use to the wider community and further increase engagement with this study. Then users can always use their own servers and imputation approaches e.g. if doing low pass sequencing.

Be good to report all results at least at the top loci in the GWAS (e.g. < 10⁻⁶). Any novel loci detected? Difficult to tell from Manhattan plots. Arguably just discussing individual, previously identified hits, is not by itself particularly informative. Especially given neither were the lead variant in

their analyses. Could for example include QQ plots pre and post imputation to see how things change genome-wide.

PHARP was published in July so shouldn't reference biorxiv paper.

The discussion is pretty limited. Don't really discuss wider context of results, new approaches (low pass sequencing etc) relative array performances etc

Line 254 should read "we applied hard filtering on them by"

Authors discuss filtering variants but was no genotype e.g. GQ filter applied? Line 271: Are there many variants left if restrict to just those with a missing rate > 0.1 across the entire cohort? I would naively assume not many if have applied GQ filters given many samples are not very high coverage. Is MAF calculated per lineage or across the whole cohort?

Reviewer #2 (Remarks to the Author):

This work has resulted in an important product that will advance swine genomics research. While many labs genotype numerous animals using various SNP chip-based products, few labs have the ability to conduct genome-level genotyping on a sufficient number of animals to conduct a thorough GWAS experiment. With the availability of the SWIM imputation server reported in this manuscript, anyone with low coverage SNP chip data can acquire genome-level genotypes on their animals for no additional charge. Furthermore, the authors have shown that their process results in the most accurate imputed genotypes based on the currently available imputation methods. To demonstrate the utility of the genotypes from their server, they conducted GWAS in two independent swine populations with different phenotypes and SNP genotypes derived from different genotyping products. Both analyses performed quite well and pointed to functional candidate SNP markers. This work represents a significant advance in swine genomics and I am certain the SWIN server will be heavily utilized by researchers around the world.

Overall the manuscript was well-written and I have very few comments that need to be addressed. One scientific question that is evident in the results is 1) Do the authors have an explanation for the slightly contradictory results presented in Figure 4 comparing PHARP to SWIM-100Y? Comparisons to Duroc and Landrace always found the SWIM concordance to be considerably higher than the respective PHARP at all levels of MAF, while for Yorkshire, the concordance values were much more similar and PHARP actually exceeded SWIM when $MAF > \sim 0.2$. This is not what I would have expected a priori and wonder if the authors know why the Yorkshire breed appears to behave slightly different from the other two breeds.

2) In the abstract, I would suggest changing 'the genetic improvement animals' to 'animal genetic improvement' in lines 28-29.

3) In line 203 change 'emails' to 'email addresses'.

4) On line 255, I think it would be easier to read if there were commas between the different criteria metrics.

5) In the figure legend for Figure 1, the sentence describing LD calculations needs to be re-worded. As currently written, it could be interpreted that all common variants ($MAF > 0.05$) were removed along with the close relatives.

Reviewer #3 (Remarks to the Author):

The manuscript presents a large reference haplotype panel and a public web server for pig genotype

imputation. It is well written, and I found it easy and enjoyable to read the manuscript. The web server is easy to use, and to my knowledge, it will be very useful for pig genetic/genomic research. I have just a few minor comments.

Minor comments

Lines 28-29: "in the genetic improvement animals" => "in animal genetic improvement"

Figure 1: i) Can add the percentage of variation explained by PC1 and PC2 in panel c; and 2) "LD was calculated with common variants (MAF >0.05), and close relatives (GRM >0.5) were removed ..."

Lines 113-125 (Figure 3): Evidence would be stronger if the analysis included a benchmark of the reference of just 250 Landrace pigs.

Line 185: "... in pigs and a recent study ..." => "... in pigs. A recent study ..."

Line 287: "All software was ..." => "All software tools were ..."

Line 316: "u is the con"?

Line 318: "... traits and, g is ..." => "... traits, and g is ..."

Line 319: What variants were used to construct the GRM? Chip SNPs, LD-pruned sequence variants, or all sequence variants? Please clarify.

Please clarify how the proportion explained by lead SNP is calculated.

The punctuation for 'respectively' should be revised.

Reviewer 1:

Q0: I thought the manuscript was generally well written with nice clear figures. My comments in no particular order are below.

Response: We appreciate the positive feedback by the reviewer.

Q1: R2 is correlated to allele frequency and Fig 1b seems to be assuming that the distribution of allele frequencies is the same for each breed. This may not be the case? For example there may be more rare variants in certain populations impacting these results which should be checked? Also is the number of animals in each group the same?

Also in the legend of Figure 1b the authors say “LD was calculated with common variants (MAF>0.05) and close relatives (GRM>0.5) removed” but in methods (line 261) say “individuals within the same breed with close relatives (GRM > 0.5) and low frequency variants (MAF < 0.05) removed”. Which seem to contradict each other with respect to whether only rare variants were kept or excluded.

Response: We thank the reviewer for this important point. r^2 is indeed highly influenced by the MAF therefore we calculated r^2 using only variants with MAF > 0.05 *within* each group. We believe this filtering step is sufficient to make the distribution of allele frequencies similar between groups (see figure below). In addition, the LD decay rates could not be explained by frequency dependent r^2 . For example, the Asian wild boars had the highest number of lower frequency variants (still > 0.05) but the lowest LD.

Number of animals for each group is included in the the legend of **Figure 1**, which has also been modified to improve clarity on SNP and animal filtering and be consistent with the description in the method section (**Line 504-506**). To be clear, low frequency variants and close relatives were removed before computing r^2 .

Q2: Figure 1D, should say “annotated above the barchart”

Response: We annotated both on top and below the barchart for (K = 6). Both are stated in the figure legend (**Line 512**). The annotation below the first barchart makes it easier to see the groupings.

Q3: Normally x axis of Figure 2/3 plots would be on a log scale, as differences are generally at the lower allele frequencies (which also generally contains most variants) e.g. see <https://www.nature.com/articles/s41588-020-00756-0/figures/2>

Response: We have changed all MAF axes to be on the log scale. This change affected Figures 2,3,4 and 3 (Figures S1-3) of the 4 additional supplemental figures.

Q4: For analysis in Figure 2 were the authors sampling the 50K SNP genotypes from the WGS data or did they have matching array data for these samples? If the former I assume not every genotype was called in the WGS data following filtering? If so should state what proportion of array genotypes were not called in the WGS data and what number were left to do the imputation with.

Response: The chip genotypes were a subset of WGS data, we did not have matching array data for these animals. This has now been clearly stated in the text (Line 109). We also indicate the number of variants where SNP chip genotypes were present in WGS data after filtering in Figure S3. Importantly, our web server outputs a run summary file that contains information regarding the number of SNPs that are overlapped with the haplotype reference panel.

Q5: Is it worth comparing imputation performance from variants on some of the other pig arrays? For example the 650k axiom array? At least for the four arrays the authors say are supported by their server here: <https://quantgenet.msu.edu/swim/statistics.php>

On line 214 they say Figure 2 was based on a 60K array but on line 109 they say was a 50K array.

Response: We thank the reviewer for this helpful suggestion. In addition to the existing analysis, which was based on GeneSeek GGP 50K array, we have now performed two new analysis based on an array with similar number of SNPs (Wens 55K) and another with a higher number of SNPs (Affy 660K). These results are summarized in a new paragraph (Line 150-160). The results are expected, higher density leads to higher imputation accuracy.

We have changed 60K to 50K in line 214 (now Line 235). The array was a 50K (GeneSeek GGP) array.

Q6: The authors suggest that for a fixed size a reference panel derived from a matching breed performs better than one composed of multiple breeds. This is contrary to the hypothesis that multi-breed reference panels better represent haplotypes that may be rare in a particular breed, improving general imputation performance. How well though does this analysis reflect likely downstream usage as 610 of the 651 landrace genomes used in this study were obtained from the same company and are therefore likely unusually related to one another. So this represents a specific sub-population of landrace animals rather than landrace as a whole. Does the imputation performance drop in the landrace animals from other cohorts?

Response: We thank the reviewer for this suggestion. This is a very important point indeed. We added a new analysis where the target set was entirely from the SRA (Figure S1-2). The imputation performance did drop, which is expected as the animals are more distantly related to animals in the reference panel. Increasing the reference size also increased accuracies. Nevertheless, we do not expect this problem to persist as more and more animals are becoming available and we update the web server.

Q7: Why is there an uptick in performance accuracies at the lowest allele frequencies in Figures 2

and 3? Normally the rarer the variant the less well imputed they are.

Response: The imputation accuracy is a complex function of many factors, including the allele frequencies, the cohort where allele frequencies are computed, the number of animals used to compute the accuracy, etc. At low frequency, r^2 may be higher due to the small number of animals used to compute the correlation even though accuracy is generally lower. However, we cannot say definitively this was the reason.

Q8: Resource spelt wrong on line 126

Response: Corrected (now Line 135).

Q9: Can the authors make the genotypes callset (VCF) for at least the 729 previously publically available and 512 new genomes available? This will likely be of substantial use to the wider community and further increase engagement with this study. Then users can always use their own servers and imputation approaches e.g. if doing low pass sequencing.

Response: The resource has been made public per the request of the reviewer (Line 364-366)

Q10: Be good to report all results at least at the top loci in the GWAS (e.g. $< 10^{-6}$). Any novel loci detected? Difficult to tell from Manhattan plots. Arguably just discussing individual, previously identified hits, is not by itself particularly informative. Especially given neither were the lead variant in their analyses. Could for example include QQ plots pre and post imputation to see how things change genome-wide.

Response: These have been added, including all results with $P < 5e-8$ (Tables S3-6) as well as QQ plots (Figure S4).

Q11: PHARP was published in July so shouldn't reference biorxiv paper.

Response: Corrected (Line 478-479).

Q12: The discussion is pretty limited. Don't really discuss wider context of results, new approaches (low pass sequencing etc) relative array performances etc

Response: We have added additional discussion points, in particular on the importance of increasing reference size, new approaches (low pass sequencing), as well as some new information on studies using the resource.

Q13: Line 254 should read "we applied hard filtering on them by"

Response: Corrected (Line 286).

Q14: Authors discuss filtering variants but was no genotype e.g. GQ filter applied? Line 271: Are there many variants left if restrict to just those with a missing rate > 0.1 across the entire cohort? I would naively assume not many if have applied GQ filters given many samples are not very high coverage. Is MAF calculated per lineage or across the whole cohort?

Response: We did not apply a GQ filter. We are in agreement with the reviewer that it would lead to fewer number of SNPs. The MAF was calculated across the whole cohort while filtering. We did apply a $>10X$ (deduplicated coverage may be lower) raw coverage filter while selecting samples, including SRA and new samples.

Reviewer 2:

Q0: This work has resulted in an important product that will advance swine genomics research. While many labs genotype numerous animals using various SNP chip-based products, few labs have the ability to conduct genome-level genotyping on a sufficient number of animals to conduct a thorough GWAS experiment. With the availability of the SWIM imputation server reported in this manuscript, anyone with low coverage SNP chip data can acquire genome-level genotypes on their animals for no additional charge. Furthermore, the authors have shown that their process results in the most accurate imputed genotypes based on the currently available imputation methods. To demonstrate the utility of the genotypes from their server, they conducted GWAS in two independent swine populations with different phenotypes and SNP genotypes derived from different genotyping products. Both analyses performed quite well and pointed to functional candidate SNP markers. This work represents a significant advance in swine genomics and I am certain the SWIN server will be heavily utilized by researchers around the world.

Response: We thank the reviewer for the insightful comments. We have revised the manuscript following the reviewer's comments.

Q1: Do the authors have an explanation for the slightly contradictory results presented in Figure 4 comparing PHARP to SWIM-100Y? Comparisons to Duroc and Landrace always found the SWIM concordance to be considerably higher than the respective PHARP at all levels of MAF, while for Yorkshire, the concordance values were much more similar and PHARP actually exceeded SWIM when $MAF > \sim .2$. This is not what I would have expected a priori and wonder if the authors know why the Yorkshire breed appears to behave slightly different from the other two breeds.

Response: Imputation accuracy is a complex function of many factors. While we are unable to offer an definitive answer, several possible explanations come to mind. First, PHARP happens to have the largest number of Yorkshires therefore haplotype representation of Yorkshires is higher than other breeds in PHARP. Second, the target set (randomly selected Yorkshires) may be genetically closer to reference haplotypes than other breeds.

Q2: In the abstract, I would suggest changing 'the genetic improvement animals' to 'animal genetic improvement' in lines 28-29.

Response: Corrected (Line 28-29).

Q3: In line 203 change 'emails' to 'email addresses'.

Response: Corrected (Line 224).

Q4: On line 255, I think it would be easier to read if there were commas between the different criteria metrics.

Response: Corrected (Line 287)

Q5: In the figure legend for Figure 1, the sentence describing LD calculations needs to be reworded. As currently written, it could be interpreted that all common variants ($MAF > 0.05$) were removed along with the close relatives.

Response: Corrected (Line 504-506).

Reviewer 3:

Q0: The manuscript presents a large reference haplotype panel and a public web server for pig genotype imputation. It is well written, and I found it easy and enjoyable to read the manuscript. The web server is easy to use, and to my knowledge, it will be very useful for pig genetic/genomic research. I have just a few minor comments.

Response: We appreciate the enthusiasm of the reviewer.

Q1: Minor comments

Lines 28-29: "in the genetic improvement animals" => "in animal genetic improvement"

Response: Corrected (Line 28-29).

Q2: Figure 1: i) Can add the percentage of variation explained by PC1 and PC2 in panel c; and 2) "LD was calculated with common variants (MAF >0.05), and close relatives (GRM >0.5) were removed ..."

Response: Corrected (Figure 1 and Line 504-506).

Q3: Lines 113-125 (Figure 3): Evidence would be stronger if the analysis included a benchmark of the reference of just 250 Landrace pigs.

Response: New analysis with only 250 Landrace pigs was performed and results added to Figure 3 (Line 122-124)

Q4: Line 185: "... in pigs and a recent study ..." => "... in pigs. A recent study ..."

Response: Corrected (Line 206).

Q5: Line 287: "All software was ..." => "All software tools were ..."

Response: Corrected (Line 319).

Q6: Line 316: "u is the con"?

Response: Corrected (Line 348).

Q7: Line 318: "... traits and, g is ..." => "... traits, and g is ..."

Response: Corrected (Line 350).

Q8: Line 319: What variants were used to construct the GRM? Chip SNPs, LD-pruned sequence variants, or all sequence variants? Please clarify.

Response: We used chip SNPs to build GRM (Line 352-353).

Q9: Please clarify how the proportion explained by lead SNP is calculated.

Response: The method to calculate variance explained is added (Line 354-355).

Q10: The punctuation for 'respectively' should be revised.

Response: We have added a comma before respectively whenever its at the end of a sentence.

REVIEWERS' COMMENTS:

Reviewer #1 (Remarks to the Author):

Thanks the authors have addressed my points.

A minor point, but regarding the response to my query to Fig 1b the authors highlight that the Asian wild boars have the highest number of low MAF variants but the most rapid drop in LD suggesting MAF doesn't confound LD. But this is actually what I would expect if MAF *was* confounding LD i.e. a more rapid drop in LD in populations with more rare variants. But the results the authors show suggest LD patterns don't seem to be exclusively explained by MAF at least.

Line 154 Probably should remove "that" in "After removal of SNPs that whose probes"

Line 159 Should this be referring to Figure S3c? Not S2c

Reviewer #2 (Remarks to the Author):

I think the authors have done a great job addressing all of the reviewer comments to this manuscript.

My only comment, which is minor, is:

Line 352 The authors need to state the complete name of the GeneSeek GGP chip. GeneSeek's first Porcine GGP chip only had ~7500 SNP. As they have previously used the name 50K GGP chip, I would suggest changing line 352 to how the chip has previously been described.

Reviewer #3 (Remarks to the Author):

The authors have responded in detail to each of the reviewers' comments. The manuscript is now in good shape for publication.

Reviewer 1:

Q1: A minor point, but regarding the response to my query to Fig 1b the authors highlight that the Asian wild boars have the highest number of low MAF variants but the most rapid drop in LD suggesting MAF doesn't confound LD. But this is actually what I would expect if MAF *was* confounding LD i.e. a more rapid drop in LD in populations with more rare variants. But the results the authors show suggest LD patterns don't seem to be exclusively explained by MAF at least.

Response: We wish to clarify that low frequency variants may lead to spurious LD thus maintaining apparent LD in the population. We agree with the reviewer that LD patterns cannot be exclusively explained by MAF.

Q2: Line 154 Probably should remove "that" in "After removal of SNPs that whose probes"

Response: Corrected.

Q3: Line 159 Should this be referring to Figure S3c? Not S2c

Response: Corrected.

Reviewer 2:

Q1: Line 352 The authors need to state the complete name of the GeneSeek GGP chip. GeneSeek's first Porcine GGP chip only had ~7500 SNP. As they have previously used the name 50K GGP chip, I would suggest changing line 352 to how the chip has previously been described.

Response: Corrected.